# PeerJ

# *dDocent*: a RADseq, variant-calling pipeline designed for population genomics of non-model organisms

Jonathan B. Puritz, Christopher M. Hollenbeck and John R. Gold

Marine Genomics Laboratory, Harte Research Institute, Texas A&M University-Corpus Christi, Corpus Christi, TX, USA

## ABSTRACT

Restriction-site associated DNA sequencing (RADseq) has become a powerful and useful approach for population genomics. Currently, no software exists that utilizes both paired-end reads from RADseq data to efficiently produce population-informative variant calls, especially for non-model organisms with large effective population sizes and high levels of genetic polymorphism. *dDocent* is an analysis pipeline with a user-friendly, command-line interface designed to process individually barcoded RADseq data (with double cut sites) into informative SNPs/Indels for population-level analyses. The pipeline, written in BASH, uses data reduction techniques and other stand-alone software packages to perform quality trimming and adapter removal, *de novo* assembly of RAD loci, read mapping, SNP and Indel calling, and baseline data filtering. Double-digest RAD data from population pairings of three different marine fishes were used to compare *dDocent* with *Stacks*, the first generally available, widely used pipeline for analysis of RADseq data. *dDocent* consistently identified more SNPs shared across greater numbers of individuals and with higher levels of coverage. This is due to the fact that *dDocent* quality trims instead of filtering, incorporates both forward and reverse reads (including reads with INDEL polymorphisms) in assembly, mapping, and SNP calling. The pipeline and a comprehensive user guide can be found at http://dDocent.wordpress.com.

Corresponding author
Jonathan B. Puritz,
jonathan.puritz@tamucc.edu

## INTRODUCTION

Next-generation sequencing (NGS) has transformed the field of genetics into genomics by providing DNA sequence data at an ever increasing rate and reduced cost (*Mardis, 2008*). The nascent field of population genomics relies on NGS coupled with laboratory methods to reproducibly reduce genome complexity to a few thousand loci. The most common approach, restriction-site associated DNA sequencing (RADseq), uses restriction endonucleases to randomly sample the genome at locations adjacent to restriction-enzyme recognition sites that, when coupled with Illumina sequencing, produces high coverage of homologous SNP (Single Nucleotide Polymorphism) loci. As such, RADseq provides a powerful method for population level genomic studies (*Ellegren, 2014*; *Narum et al., 2013*; *Rowe, Renaut & Guggisberg, 2011*).

The original RADseq approach (*Baird et al., 2008*; *Miller et al., 2007*), and initial population genomic studies employing it (*Hohenlohe et al., 2010*), focused on SNP discovery and genotyping on the first (forward) read only. This is because the original RADseq method (*Baird et al., 2008*; *Miller et al., 2007*) utilized random shearing to produce RAD loci; paired-end reads were not of uniform length or coverage, making it problematic to find SNPs at high and uniform levels of coverage across a large proportion of individuals. As a result, the most comprehensive and widely used software package for analysis of RADseq data, *Stacks* (*Catchen et al., 2013*; *Catchen et al., 2011*), provides SNP genotypes based only on first-read data. In contrast, RADseq approaches such as ddRAD (*Peterson et al., 2012*), 2bRAD (*Wang et al., 2012*), and ezRAD (*Toonen et al., 2013*) rely on restriction enzymes to define both ends of a RAD locus, largely producing RAD loci of fixed length (flRAD). Paired-end Illumina sequencing of flRAD fragments provides an opportunity to significantly expand the number of SNPs that can be genotyped from a single RADseq library.

Here, the variant-calling pipeline *dDocent* is introduced as a tool for generating population genomic data; a brief methodological outline of the analysis pipeline also is presented. *dDocent* is a wrapper script designed to take raw flRAD data and produce population informative SNP calls (SNPs that are shared across the majority of individuals and populations), taking full advantage of both paired-end reads. *dDocent* is configured for organisms with high levels of nucleotide and INDEL polymorphisms, such as are found in many marine organisms (*Guo, Zou & Wagner, 2012*; *Keever et al., 2009*; *Sodergren et al., 2006*; *Waples, 1998*; *Ward, Woodwark & Skibinski, 1994*); however, the pipeline also can be adjusted for low polymorphism species. As input, *dDocent* takes paired FASTQ files for individuals and outputs raw SNP and INDEL calls as well as filtered SNP calls in VCF format. The pipeline and a comprehensive online manual can be found at (http://dDocent. wordpress.com). Finally, results of pipeline analyses, using both *dDocent* and *Stacks*, of populations of three species of marine fishes are provided to demonstrate the utility of *dDocent* compared to *Stacks*, the first and most comprehensive, existing software package for RAD population genomics.

## METHODS

### Implementation and basic usage

The *dDocent* pipeline is written in BASH and will run using most Unix-like operating systems. *dDocent* is largely dependent on other bioinformatics software packages, taking advantage of programs designed specifically for each task of the analysis and ensuring that each modular component can be updated separately. Proper implementation depends on the correct installation of each third-party packages/tools. A full list of dependencies can be found in the user manual at (http://ddocent.wordpress.com/ddocent-pipeline-user-guide/) and a sample script to automatically download and install the packages in a Linux environment can be found at the *dDocent* repository (https://github.com/jpuritz/dDocent).

*dDocent* is run by simply switching to a directory containing input data and starting the program. There is no configuration file, and *dDocent* will proceed through a short series of command-line prompts, allowing the user to establish analysis parameters. After all required variables are configured, including an e-mail address for a completion notification, *dDocent* provides instructions on how to move the program to the background and run, undisturbed, until completion. The pipeline is designed to take advantage of multiple processing-core machines and, whenever possible, processes are invoked with multiple threads or occurrences. For most Linux distributions, the number of processing cores should be automatically detected. If *dDocent* cannot determine the number of processors, it will ask the user to input the value.

There are two distinct modules of *dDocent*: dDocent.FB and dDocent.GATK. dDocent.FB uses minimal, BAM-file preparation steps before calling SNPs and INDELs, simultaneously using FreeBayes (*Garrison & Marth, 2012*). dDocent.GATK uses GATK (*McKenna et al., 2010*) for INDEL realignment, SNP and INDEL genotyping (using HaplotypeCaller), and variant quality-score recalibration, largely following GATK Best Practices recommendations (*Van der Auwera et al., 2013*; *DePristo et al., 2011*). The modules represent two different strategies for SNP/INDEL calling that are completely independent of one another. Currently, dDocent.FB is easier to implement, substantially faster to execute, and depends on software that is commercially unrestricted; consequently, the remainder of this paper focuses on dDocent.FB. Additional information on dDocent.GATK may be found in the user guide.

## Data input requirements

*dDocent* requires demultiplexed forward and paired-end FASTQ files for every individual in the analysis (flRAD data only). A simple naming convention (a single-word locality code/name and a single-word sample identifier separated by an underscore) must be followed for every sample; examples are *LOCA_IND01.F.fq* and *LOCA_IND01.R.fq*. A sample script for using a text file containing barcodes and sample names and *process_radtags* from *Stacks* (*Catchen et al., 2013*) to properly demultiplex samples and put them in the proper *dDocent* naming convention, can be found at the *dDocent* repository (https://github.com/jpuritz/dDocent).

## Quality trimming

After *dDocent* checks that it is recognizing the proper number of samples in the current directory, it asks the user if s/he wishes to proceed with quality trimming of sequence data. If directed, *dDocent* can use the program *Trim Galore!* (http://www.bioinformatics.babraham.ac.uk/projects/trim_galore/) to simultaneously remove Illumina adapter sequences and trim ends of reads of low quality. By default, *Trim Galore!* looks for double-digest RAD adapters (*Peterson et al., 2012*) and trims bases with quality scores less than PHRED 10 (corresponding to a 10% chance of error in the base call). The read mapping and variant calling steps of *dDocent* account for base quality, so minimal trimming of the data is needed. Typically, quality trimming only needs to be performed once, so the option exists to skip this step in subsequent *dDocent* analyses.

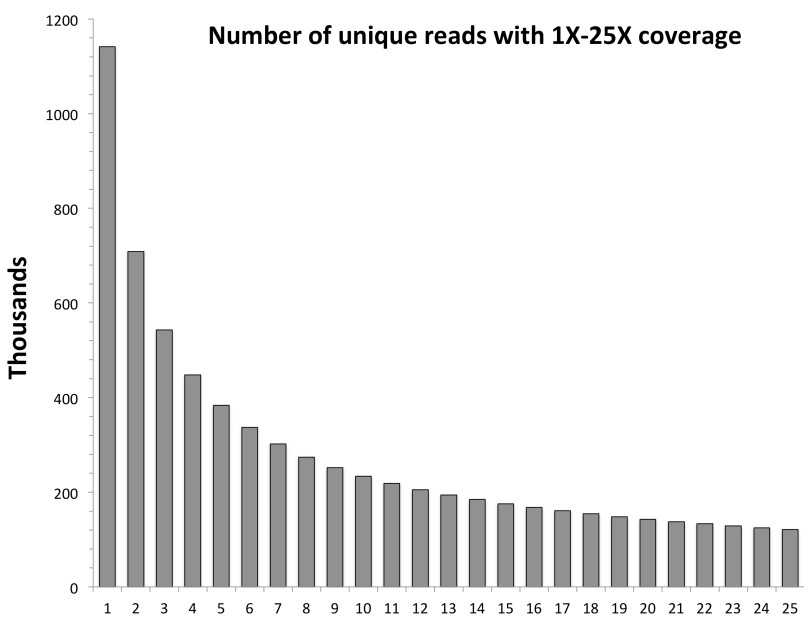

**Figure 1 Levels of coverage for each unique read in the red snapper data set.** The horizontal axis represents the minimal level of coverage, while the vertical axis represents the number of unique paired reads in thousands.

## De novo assembly

Without reference material, population genomic analyses from RADseq depend on *de novo* assembly of a set of reference contigs. Intrinsically, not all RAD loci appear in all individuals due to stochastic processes inherent in library preparation and sequencing and to polymorphism in restriction-enzyme restriction sites (*Catchen et al., 2011*). Moreover, populations can contain large levels of within-locus polymorphism, making generation of a reference sequence computationally difficult. *dDocent* minimizes the amount of data used for assembly by taking advantage of the fact that flRAD loci present in multiple individuals should have higher levels of exactly matching reads (forward and reverse) than loci that are only present in a few individuals. Caution is advised for unique reads with low levels of coverage throughout the data set as they likely represent sequencing errors or polymorphisms that are shared only by a few individuals.

In the first step of the assembly process, untrimmed, paired-end reads are reverse complemented and concatenated to forward reads. Unique paired reads are identified and their occurrences are counted in the entire data set. These data are tabulated into the number of unique reads per levels of 1X to 50X coverage; a graph is then generated and printed to the terminal. The distribution usually follows an asymptotic relationship (Fig. 1), with a large proportion of reads only having one or two occurrences, meaning they likely will not be informative on a population scale. Highly polymorphic RAD loci still should have at least one allele present at the level of expected sequence coverage, so this can be used as a guide for informative data. The user chooses a cut-off level of coverage for reads to be used for assembly—note that all reads are still used for subsequent steps of the pipeline.

After a cut-off level is chosen, remaining concatenated reads are divided back into forward- and reverse-read files and then input directly into the RADseq assembly program *Rainbow* (*Chong, Ruan & Wu, 2012*). The default parameters of *Rainbow* are used except that the maximum number of mismatches used in initial clustering is changed from four to six to help account for highly polymorphic species with large effective population sizes. In short, *Rainbow* clusters forward reads based on similarity; clusters are then recursively divided, based on reverse reads, into groups representing single alleles. Reads in merged clusters are then assembled using a greedy algorithm (*Pop & Salzberg, 2008*). *dDocent* then selects the longest contig for each cluster as the representative reference sequence for that RAD locus. If the forward read does not overlap with the reverse read (almost always the case with flRAD), the forward read is concatenated to the reverse read with ten 'N' characters as padding to represent the unknown insert. If forward and reverse reads do overlap, then a full contig is created without N padding. Finally, reference sequences are clustered based on overall sequence similarity (chosen by user, 90% by default), using the program *CD-HIT* (*Fu et al., 2012*; *Li & Godzik, 2006*). This final cluster step reduces the data set further, based on overall sequence identity after assembly. Alternatively, *de novo* assembly can be skipped and the user can provide a FASTA file with reference sequences.

## Read mapping

*dDocent* uses the MEM algorithm (*Li, 2013*) of *BWA* (*Li & Durbin, 2009*; *Li & Durbin, 2010*) to map quality-trimmed reads to the reference contigs. Users can deploy the default values of BWA or set an alternative value for each mapping parameter (match score, mismatch score, and gap-opening penalty). The default settings are meant for mapping reads to the human genome, so users are encouraged to experiment with mapping parameters. BWA output is ported to SAMtools (*Li et al., 2009*), saving disk space, and alignments are saved to the disk as binary alignment/Map (BAM). BAM files are then sorted and indexed.

## SNP and INDEL discovery and genotyping

*dDocent* uses a two-step process to optimize the computationally intensive task of SNP/INDEL calling. First, quality-trimmed forward and reverse reads are reduced to unique reads. This data set is then mapped to all reference sequences, using the previously entered mapping settings (see *Read Mapping* above). From this alignment, a set of intervals is created using BEDtools (*Quinlan & Hall, 2010*). The interval set saves computational time by directing the SNP-/INDEL-calling software to examine only reference sequences along contigs that have high quality mappings. Second, the interval list is then split into multiple files, one for each processing core, allowing SNP/INDEL calling to be optimized with a scatter-gather technique. The program *FreeBayes* (*Garrison & Marth, 2012*) is then executed multiple times simultaneously (one execution per processor and genomic interval). *FreeBayes* is a Bayesian-based, variant-detection software that uses assembled haplotype sequences to simultaneously call SNPs, INDELs, multi-nucleotide polymorphisms (MNPs), and complex events (e.g., composite insertion and substitution events) from alignment files; *FreeBayes* has the added benefit for population genomics

of using reads across multiple individuals to improve genotyping (*Garrison & Marth, 2012*). *FreeBayes* is run with minimal changes to the default parameter minimum mapping quality score and base quality score are set to PHRED 10. After all executions of *FreeBayes* are completed, raw SNP/Indel calls are concatenated into a single variant call file (VCF), using VCFtools (*Danecek et al., 2011*).

## Variant filtering

Final SNP data-set requirements are likely to be highly dependent on specific goals and aims of individual projects. To that end, *dDocent* uses *VCFtools* (*Danecek et al., 2011*) to provide only basic level filtering, mostly for run diagnostic purposes. d*Docent* produces a final VCF file that contains all SNPs, Indels, MNPs, and complex events that are called in 90% of all individuals, with a minimum quality score of 30. Users are encouraged to use VCFtools and vcflib (part of the *FreeBayes* package; https://github.com/ekg/vcflib) to fully explore and filter data appropriately.

## Comparison between *dDocent* and *Stacks*

Two sample localities, each comprising 20 individuals, were chosen randomly from unpublished RADseq data sets of three different, marine fish species: red snapper (*Lutjanus campechanus*), red drum (*Sciaenops ocellatus*), and silk snapper (*Lutjanus vivanus*). These three species are part of ongoing RADseq projects in our laboratory, and preliminary analyses indicated high levels of nucleotide polymorphisms across all populations. Double-digest RAD libraries were prepared, generally following *Peterson et al. (2012)*. Individual DNA extractions were digested with *Eco*RI and M*sp*I. A barcoded adapter was ligated to the *Eco*RI site of each fragment and a generic adapter was ligated to the *Msp*I site. Samples were then equimollarly pooled and size-selected between 350 and 400 bp, using a Qiagen Gel Extraction Kit. Final library enhancement was completed using 12 cycles of PCR, simultaneously enhancing properly ligated fragments and adding an Illumina Index for additional barcoding. Libraries were sequenced on three separate lanes of an Illumina HiSeq 2000 at the University of Texas Genomic Sequencing and Analysis Facility. Raw sequence data were archived at NCBI's Short Read Archive (SRA) under Accession SRP041032.

Demultiplexed individual reads were analyzed with *dDocent* (version 1.0), using three different levels of final reference contig clustering (90%, 96%, and 99% similarity) in an attempt to alter the most comparable analysis variable in *dDocent* to match the maximum distance between stacks parameter and the maximum distance between stacks from different individuals parameter of *Stacks*. The coverage cut-off for assembly was 12 for red snapper, 13 for red drum, and nine for silk snapper. All *dDocent* runs used mapping variables of one, three, and five for match-score value, mismatch score, and gap-opening penalty, respectively. For comparisons, complex variants were decomposed into canonical SNP and Indel representation from the raw VCF files, using *vcfallelicprimitives* from *vcflib* (https://github.com/ekg/vcflib).

For analysis with *Stacks* (version 1.08), reads were demultiplexed and cleaned using *process_radtags*, removing reads with 'N' calls and low-quality base scores. Because *dDocent* inherently uses both reads for SNP/Indel genotyping, forward reads and reverse reads

were processed separately with *denovo_map.pl*, using three different sets of parameters. The first set had a minimum depth of coverage of two to create a stack, a maximum distance of two between stacks, and a maximum distance of four between stacks from different individuals, with both the deleveraging algorithm and removal algorithms enabled. The second set had a minimum depth of coverage of three to create a stack, a maximum distance of four between stacks, and a maximum distance of eight between stacks from different individuals, with both the deleveraging algorithm and removal algorithms enabled. The third set had a minimum depth of coverage of three to create a stack, a maximum distance of four between stacks, and a maximum distance of 10 between stacks from different individuals, with both the deleveraging algorithm and removal algorithms enabled. SNP calls were output in VCF format.

For both *dDocent* and *Stacks* runs, VCFtools was used to filter out all INDELs and SNPs that had a minor allele count of less than five. SNP calls were then evaluated at different individual-coverage levels: the total number of SNPs; the number of SNPs called in 75%, 90%, and 99% of individuals at 3X coverage; the number of SNPs called in 75% and 90% of individuals at 5X coverage; the number of SNPs called in 75% and 90% of individuals at 10X coverage; and the number of SNPS called in 75% and 90% of individuals at 20X coverage. Overall coverage levels for red snapper were lower and likely impacted by a few low-quality individuals; consequently, the number of 5X and 10X SNPs shared among 90% of individuals (after removing the bottom 10% of individuals in terms of coverage) were compared instead of SNP loci shared at 20X coverage. Results from two runs of *Stacks* (one using forward and one using reverse reads) were combined for comparison with *dDocent*, which inherently calls SNPs on both reads. All analyses and computations were performed on a 32-core Linux workstation with 128 GB of RAM.

## RESULTS AND DISCUSSION

Results of SNP calling, including run times (in minutes) for each analysis (not including quality trimming), are presented in Table 1. Data from high coverage SNP calls, averaged over all runs for each pipeline, are presented in Fig. 2. While *Stacks* called a larger number of low coverage SNPs, limiting results to higher individual coverage and to higher individual call rates revealed that *dDocent* consistently called more high-quality SNPs. Run times were equivalent for both pipelines.

At almost all levels of coverage in three different data sets, *dDocent* called more SNPs across more individuals than *Stacks*. Two key differences between *dDocent* and *Stacks* likely contribute these discrepancies: (i) quality trimming instead of quality filtering, and (ii) simultaneous use of forward and reverse reads by *dDocent* in assembly, mapping, and genotyping, instead of clustering as employed by *Stacks*. As with any data analysis, quality of data output is directly linked to the quality of data input. Both *dDocent* and *Stacks* use procedures to ensure that only high-quality sequence data are retained; however, *Stacks* removes an entire read when a sliding window of bases drops below a preset quality score (PHRED 10, by default), while *dDocent* via *Trim Galore!* trims off low-quality bases, preserving high-quality bases of each read. Filtering instead of trimming results in fewer

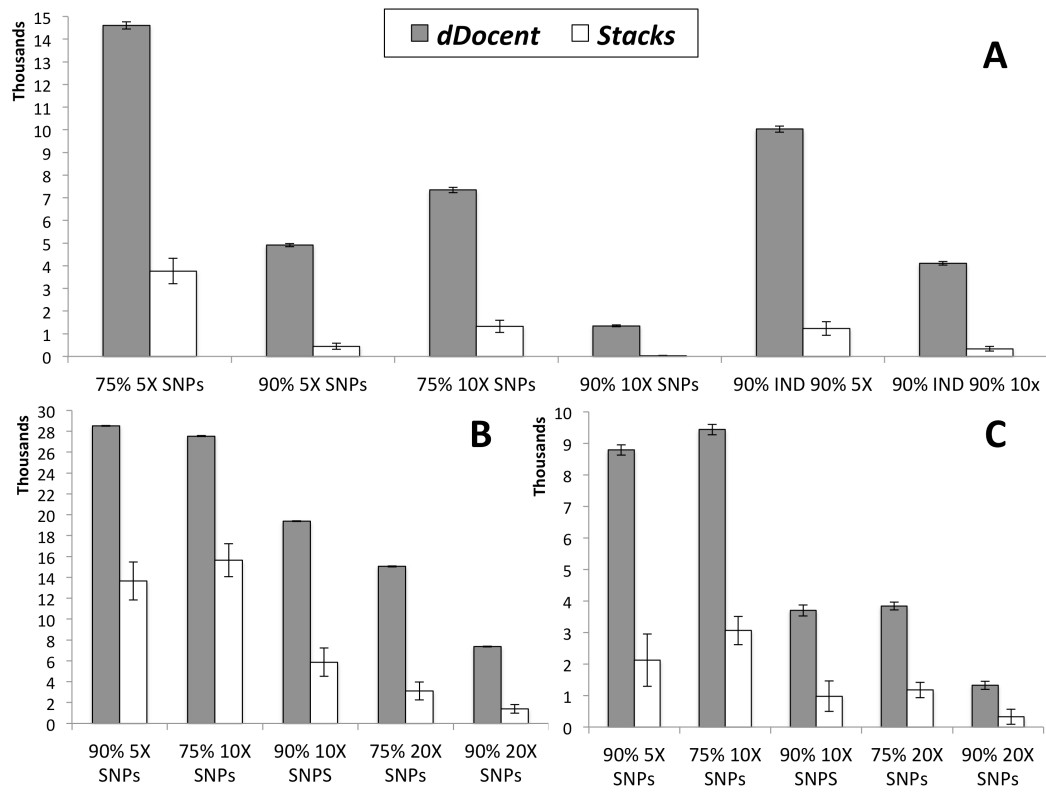

**Figure 2 SNP results averaged across the three different run parameters for *dDocent* and *Stacks*.** (A) Red snapper, (B) Red drum, (C) Silk snapper (see Methods or Table 1 for SNP categories description). Error bars represent one standard error.

reads entering the *Stacks* analysis (between 65% and 95% of the data compared to *dDocent*; data not shown), generating lower levels of coverage and fewer SNP calls than *dDocent*.

*dDocent* offers two advantages over *Stacks*: (i) it is specifically designed for paired-end data and utilizes both forward and reverse reads for *de novo* RAD loci assembly, read mapping, variant discovery, and genotyping; and (ii) it aligns reads to reference sequence instead of clustering by identity. Using both reads to cluster and assemble RAD loci helps to ensure that portions of the genome with complex mutational events, including INDELS or small repetitive regions, are properly assembled and clustered as homologous loci. Additionally, using *BWA* to map reads to reference loci enables *dDocent* to properly align reads with INDEL polymorphisms, increasing coverage and subsequent variant discovery and genotyping. Clustering methods employed by *Stacks*, whether clustering alleles within an individual or clustering loci between individuals, effectively remove reads, alleles, and loci with INDEL polymorphisms because the associated frame shift effectively inflates the observed number of base-pair differences. For organisms with large effective population sizes and high levels of genetic diversity, such as many marine organisms (*Waples, 1998*; *Ward, Woodwark & Skibinski, 1994*), removing reads and loci with INDEL polymorphisms will result in a loss of shared loci and coverage.

**Table 1 Results from individual runs of *dDocent* and *Stacks*.** *dDocent* runs varied in the level of similarity used to cluster reference sequences: A (90%), B (96%), and C (99%). For *Stacks*, forward reads and reverse reads were separately processed with *denovo_map.pl* (*Stacks* version 1.08), using three different sets of parameters: A, minimum depth of coverage of two to create a stack, a maximum distance of two between stacks, and a maximum distance of four between stacks from different individuals; B, minimum depth of coverage of three to create a stack, a maximum distance of four between stacks, and a maximum distance of eight between stacks from different individuals; and C, minimum depth of coverage of three to create a stack, a maximum distance of four between stacks, and a maximum distance of 10 between stacks from different individuals. For *dDocent*, complex variants were decomposed into canonical SNP and INDEL calls and INDEL calls were filtered out. SNP calls were evaluated at different individual coverage levels: (i) total number of SNPs; (ii) number of SNPS called in 75%, 90%, and 99% at 3X coverage; (iii) number of SNPS called in 75% and 90% of individuals at 5X coverage; (iv) number of SNPS called in 75% and 90% of individuals at 10X coverage; and, (v) number of SNPS called in 75% and 90% of individuals at 20X coverage. Run times are in minutes. Results from forward and reverse reads of *Stacks* were combined for comparison with *dDocent*, which inherently calls SNPs on both reads.

| | *dDocent* A | *dDocent* B | *dDocent* C | *Stacks* A | *Stacks* B | *Stacks* C |
|---|---|---|---|---|---|---|
| | | | **Red snapper** | | | |
| Total 3X SNPS | 53,298 | 53,316 | 53,361 | 28,817 | 33,479 | 34,459 |
| 75% 3X SNPs | 21,195 | 20,990 | 20,724 | 4,150 | 5,735 | 5,728 |
| 90% 3X SNPs | 9,102 | 8,850 | 8,639 | 675 | 987 | 983 |
| 99% 3X SNPs | 78 | 47 | 15 | – | – | – |
| 75% 5X SNPs | 14,881 | 14,594 | 14,339 | 2,632 | 4,351 | 4,324 |
| 90% 5X SNPs | 5,021 | 4,925 | 4,785 | 179 | 579 | 574 |
| 75% 10X SNPs | 7,556 | 7,318 | 7,154 | 783 | 1,618 | 1,579 |
| 90% 10X SNPS | 1,414 | 1,340 | 1,286 | 7 | 48 | 47 |
| 90% IND 90% 5X | 10,267 | 10,026 | 9,798 | 806 | 1,807 | 1,079 |
| 90% IND 90% 10x | 4,242 | 4,093 | 3,974 | 129 | 441 | 434 |
| Run time | 41 | 41 | 42 | 70 | 47 | 53 |
| | | | **Red drum** | | | |
| Total 3X SNPS | 46,378 | 46,688 | 46,832 | 45,792 | 50,821 | 52,366 |
| 75% 3X SNPs | 36,745 | 36,905 | 36,900 | 24,134 | 28,991 | 28,981 |
| 90% 3X SNPs | 32,356 | 32,424 | 32,330 | 13,439 | 17,946 | 17,874 |
| 99% 3X SNPs | 11,906 | 11,910 | 11,774 | 828 | 1,264 | 1,259 |
| 75% 5X SNPs | 34,279 | 34,393 | 34,336 | 21,021 | 26,526 | 26,464 |
| 90% 5X SNPs | 28,532 | 28,566 | 28,431 | 10,494 | 15,282 | 15,207 |
| 75% 10X SNPs | 27,523 | 27,605 | 27,488 | 12,928 | 17,018 | 16,983 |
| 90% 10X SNPS | 19,434 | 19,442 | 19,283 | 4,159 | 6,734 | 6,705 |
| 75% 20X SNPs | 15,080 | 15,111 | 14,981 | 2,276 | 3,538 | 3,516 |
| 90% 20X SNPs | 7,365 | 7,409 | 7,304 | 243 | 1,974 | 1,961 |
| Run time | 43 | 45 | 45 | 58 | 55 | 65 |
| | | | **Silk snapper** | | | |
| Total 3X SNPS | 68,668 | 68,825 | 68,861 | 48,742 | 55,505 | 58,352 |
| 75% 3X SNPs | 30,771 | 30,391 | 30,051 | 7,596 | 9,705 | 9,696 |
| 90% 3X SNPs | 14,952 | 14,673 | 14,415 | 2,007 | 3,439 | 3,433 |
| 99% 3X SNPs | 4,294 | 4,060 | 3,952 | 132 | 527 | 523 |
| 75% 5X SNPs | 20,534 | 20,188 | 19,968 | 4,789 | 7,290 | 7,274 |

Table 1 (*continued*)

|  | *dDocent* A | *dDocent* B | *dDocent* C | *Stacks* A | *Stacks* B | *Stacks* C |
|---|---|---|---|---|---|---|
| 90% 5X SNPs | 9,103 | 8,750 | 8,533 | 1,225 | 2,573 | 2,570 |
| 75% 10X SNPs | 9,765 | 9,400 | 9,159 | 2,094 | 3,547 | 3,546 |
| 90% 10X SNPS | 3,923 | 3,691 | 3,490 | 489 | 1,224 | 1,223 |
| 75% 20X SNPs | 4,069 | 3,832 | 3,624 | 703 | 1,415 | 1,411 |
| 90% 20X SNPs | 1,431 | 1,313 | 1,228 | 136 | 417 | 418 |
| Run time | 88 | 95 | 59 | 93 | 89 | 204 |

    *dDocent* is specifically designed to efficiently generate SNP and INDEL polymorphisms that are shared across multiple individuals. To that end, the output reference contigs and variant calls represent a subset of the total, genomic information content of the raw input data; RAD loci and variants present in single individuals are largely ignored. Other analysis software, such as the scripts published by *Peterson et al. (2012)*, represent a more comprehensive alternative for generating for a full *de novo* assembly of RAD loci and would increase the chance of discovering individual level polymorphisms. For population genomics, loci that are not shared by at least 50% of all individuals and/or have minor allele frequencies of less than 5% are often filtered out. *dDocent* saves computational time by ignoring these loci from the outset of assembly; however, users can pass in a more comprehensive reference (including an entire genome) in order to include all possible variant calls from the data.

## CONCLUSION

*dDocent* is an open-source, freely available population genomics pipeline configured for species with high levels of nucleotide and INDEL polymorphisms, such as many marine organisms. The *dDocent* pipeline reports more SNPs shared across greater numbers of individuals and with higher levels of coverage than current alternatives. The pipeline and a comprehensive online manual can be found at (http://dDocent.wordpress.com) and (https://github.com/jpuritz/dDocent).

## ACKNOWLEDGEMENTS

We thank T Krabbenhoft for assistance in beta testing, and C Bird and D Portnoy for useful discussions and comments on the manuscript. We also would like to thank the three reviewers for their substantial help with troubleshooting the user guide and installation process on multiple computing platforms.

### Funding

This work was supported by Award #NA10NMF4270199 from the Saltonstall-Kennedy Program of the National Marine Fisheries Service (Department of Commerce/National Oceanic and Atmospheric Administration), Award #NA12NMF4330093 from the Marfin Program of the National Marine Fisheries Service, Award #NA12NMF4540082 from

the Cooperative Research Program of the National Marine Fisheries Service, and Award # NA10OAR4170099 from the National Oceanic and Atmospheric Administration to Texas Sea Grant, and by TexasAgriLife under Project H-6703. The statements, findings, conclusions, and recommendations are those of the author(s) and do not necessarily reflect the views of the National Marine Fisheries Service, the National Oceanic and Atmospheric Administration (NOAA), the U.S. Department of Commerce, Texas Sea Grant, or Texas AgriLife. The funders had no role in study design, data collection and analysis, decision to publish, or preparation of the manuscript.

### Grant Disclosures

The following grant information was disclosed by the authors:
National Marine Fisheries Service under Saltonstall-Kennedy Award: NA10NMF4270199.
MARFIN Award: NA12NMF4330093.
Cooperative Research Program Award: NA12NMF4540082.

### Competing Interests

The authors declare there are no competing interests.

### Author Contributions

- Jonathan B. Puritz conceived and designed the experiments, performed the experiments, analyzed the data, contributed reagents/materials/analysis tools, wrote the paper, prepared figures and/or tables, reviewed drafts of the paper.
- Christopher M. Hollenbeck performed the experiments, contributed reagents/materials/analysis tools, wrote the paper, prepared figures and/or tables, reviewed drafts of the paper.
- John R. Gold contributed reagents/materials/analysis tools, wrote the paper, prepared figures and/or tables, reviewed drafts of the paper.

### DNA Deposition

The following information was supplied regarding the deposition of DNA sequences:
Raw reads were deposited to the Short Read Archive of NCBI (Accession SRP041032).

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
