# Peer review of "dDocent: a RADseq, variant-calling pipeline designed for population genomics of non-model organisms"

_PeerJ, doi:10.7717/peerj.431_

## Round 0.1 · original submission · Minor Revisions

I am sorry that the reviewing process took longer than anticipated. On the flip side, as you will see the reviewers have taken their time to carefully test the pipeline and evaluate your manuscript in an open peer review process, and the feedback they have given is generally very positive.

However, the reviewers have also highlighted some issues, with their most important criticisms broadly relating to the following:

- The pipeline installation process is not straightforward and would benefit from a more comprehensive documentation and potentially some minor amendments to the pipeline itself.
- Datasets used for comparison with STACKS need to be publicly deposited and their IDs referenced in the paper, to make it straightforward to reproduce the analysis reported in the manuscript.
- It would be helpful to provide a more detailed explanation of how the default parameter settings for individual pipeline components were chosen, as well as potentially implement a simpler way to supply custom parameter settings.
- Another relevant SNP-calling tool, ddRAD, needs to be given some attention in the manuscript.

·

Basic reporting

I greatly enjoyed reading the MS by Puritz et al. The new pipeline described in the manuscript provides a welcomed new analytic approach for the analysis of Rad-Seq data. The MS is both straightforward and well written. I really liked the fact that the new pipeline capitalizes on a number of existing and well tried scripts for Rad-Seq. I have annotated a few comments in the pdf draft that the authors may want to consider.

While the MS is self-contained, it does require that readers also check the relevant Web site with additional relevant information. The Web site is well laid out and easy to follow.

Experimental design

The new pipeline is elegantly described in the MS. My only (perhaps more significant comment) is related to the intended target audience. I had no problems installing the pipeline and its dependencies in my Linux based system following basic instructions available in the author’s web site. I argue, however, that even with the instructions provided, potential interested users not familiar with Linux may struggle a bit. Please see below (General Comments for the Author) for suggestions.

Validity of the findings

The summary results, presented in the relevant table and figures, are very straightforward and easy to follow. It is clear that the new pipeline has a number of useful advantages. I find it somewhat frustrating, however, that the authors have not provided a demo data set to allow potential users to try the pipeline. The idea is that users should be able to reproduce results from a known data set before trying their own data.

Naturally, I appreciate the logistics involved given the volume of data involved and potential implications of disclosing (possibly) unpublished data. I wonder, however, if the authors couldn't prepare a smaller data set that could provide potential users with a valuable learning data set. Even better if the authors lead users through the results.

Additional comments

Maybe the authors would like to consider including a step by step tutorial (on their Web site) with specific “digested” instructions of how installing the pipeline in their systems. This would include setting of the PATH variable (on their own Web site rather than pointing users to another link) and the need to have both “git” and “curl” previously installed in the Linux system. This is certainly the case for Ubuntu.

While I appreciate that this will not to be a problem for accomplished Linux users, these days it is pretty straightforward to gain access to a Linux system even for users more familiar with other OS. In my experience, users (in particular those with limited experience or just starting) will always start with the easiest solution.

If the author go through the hassle to also consider “newbies”, I strongly believe they will benefit later on in terms of “citations”/support to their pipeline. Please note this is only a suggestion. I can appreciate how fast this can escalate in terms of time and work involved.

Another very minor point; it’s not very clear in the text (if it is, I missed it) whether the new pipeline will also work on data previously generated by original Rad-seq protocol (Baird et al., 2008). Maybe this could be clarified further. Otherwise, a very interesting and potentially useful MS, which certainly has my recommendation for publication!

·

Basic reporting

Basic reporting meets the PeerJ standards, though I have made a few grammatical suggestions in "General Comments". Introduction, discussion, tables and figures are appropriate to the topic and relevant literature is cited.

Experimental design

The article describes a software pipeline that brings together a series of cutting-edge bioinformatics programs to lower the activation energy required for empirical population geneticists to use restriction-site associated DNA (RAD) data in their research. The authors test the pipeline on ddRAD data from three species of marine fish and show that it consistently calls more polymorphisms than the most popular pipeline that is currently used in our field.

Validity of the findings

I completed a brief test of the authors' claims that dDocent is able to call more polymorphisms than Stacks by using a small ddRAD dataset of 8 kelp rockfish from Carmel Bay (Sebastes atrovirens, ~18M reads, paired end 2x300, cut with EcoR1 and Sph1, size selected at 400bp) I ran dDocent with a clustering parameter of 0.99 (~4 bases out of our mean aligned read length of 350bp) to bring it as close as possible to the maximum graph distance of 2 between "stacks" that I had used for Stacks and default bwa alignment parameters. I chose to keep unique reads with a coverage depth of 15 based on the the graph of coverage depth that it showed me.

I compared output from this run to a Stacks v1.12 run with minimum stack depth of 4, max distance of 2 between stacks, and max distance of 2 between stacks from different individuals. I output SNPs from Stacks with a minimum read depth of 5,10,15 and 20 and that occurred within >85% (7/8) of the sampled individuals to make this near equivalent with the 0.90 threshold of dDocent.

The run took about 4 hours which was similar to what I recall for the Stacks run. At 15x coverage dDocent found 927 polymorphisms at 533 loci, 770 of which were SNPs (the rest being indels or more complex variants). At this same level of coverage, Stacks found 45 SNPs at 22 loci, with no information about other variants. Even at 10x coverage, Stacks found only 465 SNPs at 229 loci, which still underperformed dDocent. Only at 5x coverage did Stacks find more SNPs.

This test validates the findings presented by the authors in Table 1 and Figure 2. I would only ask that they confirm and clarify that the numbers that they give from dDocent pipeline are for SNPs and not for all polymorphisms so that we are comparing apples to apples.

Additional comments

Until now, Stacks has been by far the most popular program for processing data from RAD methods due to its ease of use, early availability and the (especially) the high activation energy required to learn more sophisticated modular programs necessary for a SNP-calling pipeline. By bringing together a series of cutting-edge bioinformatics programs to make SNP calls the dDocent pipeline can help to lower this barrier for non-bioinformaticians.

My major comments have to do with usability, compatibility and reproducibility:

1) Instructions for installing the program should be more clear. I worked with the corresponding author to install the pipeline on our Mac Pro and found numerous incompatibilities with OSX 10.9.2. This is not the authors' fault but he was able to fix this for future users. We also identified a few other issues that may result from unclear instructions on the website.
a) The installation instructions state that executables should be in your $PATH directory. However, it is possible to have more than one directory on one's $PATH, so the instructions should clarify that all executables should be in the SAME directory.
b) The program was designed for 2×100 paired end reads, while our lab is getting 2×300 from our MiSeq. The corresponding author modified the script to measure sequence length.
c) Two dependencies, bedtools and bamtools, were required for the pipeline, but not listed on the website.
d) When I installed vcftools, I found that I still needed to manually move the Vcf.pm perl module to the perl directory. This issue may be platform specific.

2) While it is good that dDocent lowers the threshold to bioinformatic processing, the authors should be careful not to lower it too much. Because the pipeline allows SNPs to be called with the push of a few buttons, I can envision numerous users blindly accepting the output even when default parameter settings are not optimal for their system. I would recommend that the authors:

a) provide better access to the parameter settings of each component of the pipeline. Customization is allowed for sequence clustering and mapping, but what about trimming, SNP calling and variant filtering? It should not be difficult to pass these parameters into the script. I would recommend that the user fill out a text file specifying each of these, rather than have a lengthy set of questions at the beginning of each run. This text file would also be useful for documenting the settings that the user used for a particular run - important for reproducibility.

b) provide a better explanation of the settings for each step in the process, on the website and in the downloadable package, similar to what is done on the Stacks website. Of course it is the user's responsibility to make scientifically sound decisions here, but too many will be happy to accept the defaults - which could lead to a poor reputation for the pipeline.


Specific Comments:

Abstract:
Sentence starting with "Currently" is a bit of a run-on. Is the software uniquely applicable only to organisms with large Ne or is it more that it is unique in its application to all RADseq data?

Introduction:
L32: please clarify "population informative"
L33: "dDocent is configured for organisms with high levels of nucleotide and indel polymorphisms such as found in many marine organisms" - how so? please describe here or below, but within in the introduction. Could the configuration be changed to deal with less genetically diverse species?

L34: such as ARE found

L47: Installation instructions should be clearer on the website. See above.

L58: "Should be" or "are" invoked?

L63: I think I understand why the authors are focusing on the FreeBayes version of the script, but a better explanation is needed for PeerJ readers, or else it might make sense to remove mention of the GATK version from the manuscript?

L78: The script allows customization of program parameters for BWA and CD-hit. Why not allow customization of the parameters in Trim Galore?

L98: I hope the paired end reads are reverse complemented and aligned, rather than concatenated? How is this achieved? (With what algorithm?)

L98: Providing a graph of coverage and allowing the user to choose expected coverage for the assembly is a creative approach to the coverage problem. I like it.

L110: Why not allow the user to set the program parameters for Rainbow as well?

L111: should be, or are?

L115: same question as above... if the reads do overlap, are they aligned to one another and how? In our lab, the forward and reverse reads almost always overlap, and my impression from Peterson et al. 2013, is that the reads are supposed to overlap for ddRAD.

L137 - maybe easier to understand if you say something like "split into multiple files, one for each processing core"

L145 - again, it would be good to let the user have access to change these parameters

L153 - and again, it would be nice to have control over this parameter

L157 - what dDocent version was used, and what Stacks version was used in this comparison (both programs are evolving)?

L158 - "comprising", not "comprised of"

L173 - what analysis parameters are you trying to match in stacks? M (Maximum distance between stacks?)? Be clear.

223 - I think it would be good to add a third point here about the advantages that Freebayes brings over both Stacks and GATK, like haplotype awareness, lack of post-alignment processing and speed.

·

Basic reporting

The article is clearly written and represents a sound pipeline for ddRAD data analysis. The basic reporting criteria are met. In terms of the PeerJ policies there is no clear description/plan for deposition of the test datasets in a publically available repository. As such, it is not possible for anyone to recreate the data analyses presented in the paper specifically, only the pipeline installation running. I recommend depositing the test data in NCBI/EMBL or similar prior to publication so that reproduciblity/verification of the efficiency reported here is possible.

Experimental design

The submission appears rigorous, clear, and timely. Aside from the lack of access to the comparison datasets analyzed herein, the methods appear sufficiently reproducible (though see specific bug fixes mentioned below). All other experimental procedures conform to the PeerJ guidelines/evaluation criteria.

Note these issues below are when trying to install the pipeline on a Dell Precision T7610 running Linux RedHat Enterprise version 6.
There are a number of problems with the install requirements script posted on the authors GitHub repository (install_dDocent.FB_requirements).
1-Gnuplot is a dependency not included in the install script
2-tar doesn't recognize gzip format of the ch-hit link
3-freebayes didn't make via automated script
4-vcftools didn't make either
#need to update PERL5LIB as per vcftools instructions export PERL5LIB=/home/labcomp/programs/dDocent/vcftools_0.1.11/perl/

5-It is not clear what data are being used for the assembly as the quality trimming step happens alongside other dDocent processes. Does the pipeline use untrimmed reads for the assembly (clearer description of this should be added to the methods)?
6-Running the pipeline without trimming fails on the mapping/bwa-mem step which appears to be a filename issue:
[E::main_mem] fail to open file `LLM_282.R1.fq'.

7-Put units of run time in the table 1 legend.

8-my version of cd-hit-est did not have a -mask option so had to disable in order for this step to function (it simply doesn't run cd-hit-est with the -mask N option added in)
#cd-hit-est -i rainbow.fasta -o reference.fasta -mask N -M 0 -T 0 -c $simC &>cdhit.log
cd-hit-est -i rainbow.fasta -o reference.fasta -M 0 -T 0 -c $simC &>cdhit.log

Validity of the findings

The pipeline appears robust, however, the authors did not deposit the data used for the comparison hence I cannot evaluate the validity of the test findings specifically. The conclusions are appropriately stated and the presented results accurately highlight the advantages/disadvantages of the dDocent pipeline compared to STACKS. It is interesting to not that the authors do not mention the ddRAD analysis pipeline of Peterson et al. 2012, only the library prep method.
I might suggest also adding some text (no need for a full fledged analysis comparison) at least describing the various aspects of the Peterson pipeline and how dDocent differs, and the pros and cons of both strategies. One difference that is immediately apparent is the data exclusion step of dDocent based on unique sequence coverage. See below for an excerpt from Peterson et al which calls into question simple exclusion of low coverage unique sequences and/or singletons. Some discussion of this aspect should be included.

"Previous RADseq unreferenced analyses have employed a variety of heuristic approaches to distinguish among these categories, such as discarding singleton reads to eliminate error-containing reads, grouping sequences that differ by 1–3 mismatches to identify sets of homologous alleles, and discarding homolog sets consisting of unusually large numbers of reads to eliminate paralog and interspersed repeats [10], [12]. These approaches are both inefficient (it is likely that an error that generates a singleton will occur at a non-polymorphic site and as such, the majority of error-containing reads are still informative) and arbitrarily restrictive, as insertions/deletions (indels), polymorphisms and multiple SNP haplotypes require extension beyond single-mismatch homology."

It is clear the authors of this submission were interested in minimizing computation time, however some sacrifices are made by streamlining the data and caution should be advised to the readers about where these sacrifices might manifest themselves.

Additional comments

Overall the submission represents a powerful, streamlined analytical pipeline for the analysis of ddRAD data. There are a number of issues with the dependency installation script (a noble idea but perhaps it can be modified to simply instruct the user on how to install things instead of trying to automate the entire process, that way the most current versions could be downloaded instead of outdated ones as time progresses). Additional discussion of the analytical pipeline of Peterson et al 2012 should be included to highlight the pros and cons of this more streamlined and user-friendly pipeline versus the more complicated and computationally intensive pipeline presented by Peterson. Also, all data analysed for this submission should be made publically available prior to publication.

---

## Round 0.2 · accepted · Accept

Thank you very much for addressing the reviewers' and my comments, particularly regarding the pipeline documentation, but also making changes to the manuscript and the pipeline itself. I am now happy to accept your manuscript for publication in PeerJ.